# Batch Normalization Is Sufficient for Universal Function Approximation in CNNs

**Rebekka Burkholz**
CISPA Helmholtz Center for Information Security
66123 Saarbrücken, Germany
`burkholz@cispa.de`

## Abstract

Normalization techniques, for which Batch Normalization (BN) is a popular choice, is an integral part of many deep learning architectures and contributes significantly to the learning success. We provide a partial explanation for this phenomenon by proving that training normalization parameters alone is already sufficient for universal function approximation if the number of available, potentially random features matches or exceeds the weight parameters of the target networks that can be expressed. Our bound on the number of required features does not only improve on a recent result for fully-connected feed-forward architectures but also applies to CNNs with and without residual connections and almost arbitrary activation functions (which include ReLUs). Our explicit construction of a given target network solves a depth-width trade-off that is driven by architectural constraints and can explain why switching off entire neurons can have representational benefits, as has been observed empirically. To validate our theory, we explicitly match target networks that outperform experimentally obtained networks with trained BN parameters by utilizing a sufficient number of random features.

## 1 Introduction

Normalization techniques (Salimans & Kingma, 2016; Huang et al., 2017), for which Batch Normalization (BN) (Ioffe & Szegedy, 2015) is a popular choice, is an integral part of many deep learning architectures and contributes significantly to the learning success of CNNs. It has been designed to stabilize and accelerate the training process by normalizing intermediate feature maps during mini-batch processing. As it also enables the use of larger learning rates, it often improves the generalization performance (Bjorck et al., 2018). Effectively, it learns affine linear transformations of neurons. Formally, these transformations are redundant, as they could be integrated in the weights and biases of the preceding layer. For that reason, theoretical insights that try to explain the significant impact of BN on generalization usually focus on its role for the training dynamics (Bjorck et al., 2018; Santurkar et al., 2018).

Yet Frankle et al. (2021) have observed an intriguing empirical phenomenon that suggests that batch normalization can still contribute substantially to the expressiveness of CNNs. Removing random BN parameters from the network impacts the performance of a trained model more severely than the removal of random parameters. This is in line with the finding that BN parameters are particularly effective in finetuning language models Ben Zaken et al. (2022); Lu et al. (2022). Furthermore, training BN parameters alone while keeping the remaining neural network parameters frozen to their initial values can achieve nontrivial performance on standard image classification benchmark data, especially, if the network with or depth are increased (Frankle et al., 2021).

To explain these observations, we delve into the role of normalization as a fundamental technique for universal function approximation within CNNs. We prove rigorously that training normalization layers alone is sufficient to represent a wide spectrum of target layers, as long as the underlying architecture is sufficiently wide and deep. In doing so, we derive precise requirements on the width, depth, and their trade-off for convolutional and residual architectures and a diverse set of activation functions. Our insights imply improvements of bounds that have been established for fully connected networks (Giannou et al., 2023).

Utilizing our explicit constructions of target layers, we further provide an empirical proof of concept that random CNNs with learned normalization layers can compete with targets that are fully trained, which could previously not be achieved by training normalization layers from scratch.

Our results thus establish that normalization layers contribute to the expressiveness of CNNs by linearly combining features. They furthermore lay the foundation for investigating to which degree we could save memory and computational costs by keeping a high proportion of neural networks fixed to random values.

**Contributions** 1) We prove that modern convolutional architectures with randomly initialized tensors are fully expressive if they are wide and deep enough and only the normalization layers are trained. Our results state precise width requirements to represent target networks of a given size. 2) These improve previous bounds for fully-connected networks (Giannou et al., 2023). 3) Our theorems apply to a large class of activation functions including ReLUs, LeakyReLUs, tanh, and sigmoids. 4) Our experiments on CIFAR10 and CIFAR100 provide an empirical proof of concept that training only normalization layers can achieve competitive performance.

## 1.1 RELATED WORK

**Insights into BN** BN has been established as default component of most modern neural network architectures due to its significant benefit on training speed and generalization, as it enables training with larger learning rates (Bjorck et al., 2018). It makes the training success robust to different choices of parameter initializations, as it improves initial signal propagation (De & Smith, 2020; Joudaki et al., 2023), and smoothens the loss landscape (Santurkar et al., 2018). An interesting mechanism how it contributes to effective learning is the orthogonalization of features (Daneshmand et al., 2021) in combination with learning their affine transformations.

**Training only BN** (Giannou et al., 2023) is closest to our work as they derive bounds for the width of fully-connected network layers with ReLUs that are affine linearly transformed to represent an arbitrary target layer. In comparison, our results apply to convolutional structures (which encompass fully-connected layers as a special case), apply to a large class of activation functions, and improve the bounds for the construction of a target layer from two source layers in the fully-connected setting. Furthermore, we derive an novel construction that utilizes multiple source layers. It is based on CNNs (instead of residual targets) and achieves a much lower bound on the required target width. This work was inspired by (Frankle et al., 2021), who propose the conjecture that training BN layers might be fully expressive based on experimental insights for CNNs and ResNets. While (Rosenfeld & Tsotsos, 2019) have also studied training only BN parameter experimentally, they could not reach the same conclusion, as they have focused on relatively short training periods.

**Alternatives to BN** The high computational and memory costs of BN and the fact that it breaks the independence of minibatch samples and prevents adversarial training (Wang et al., 2022) are all major disadvantages, which have inspired the search for alternatives Zhang et al. (2018); Brock et al. (2021b). A combination of scaled weight standardization and gradient clipping has recently outperformed BN (Brock et al., 2021b). Our derivations for general affine transformations and different normalization mechanisms also apply to this setting. Otherwise, initialization approaches that are tailored to specific architectures can also make BN obsolete (Balduzzi et al., 2017; Burkholz & Dubatovka, 2019; De & Smith, 2020; Zhang et al., 2018; Gadhikar & Burkholz, 2022). The generation of such random parameters might also induce an inductive bias that could induce fewer affine transformation parameters.

**Lottery Ticket Existence** Our theoretical set-up is similar to lottery ticket existence proofs in fully-connected layers (Orseau et al., 2020; Pensia et al., 2020; Burkholz, 2022b) and CNNs (da Cunha et al., 2022; Burkholz, 2022a; Burkholz et al., 2022) in the sense that we construct targets based on two or more (Burkholz, 2022b; Gadhikar et al., 2023) random source network layers. Existence results in this context are also concerned with the question of universality versus specificity to targets that could reduce the width requirements (Burkholz et al., 2022) and solve trade-offs related to the depth and width of a target network representation. The underlying questions are fundamentally related, as the search for strong lottery tickets extracts information from random source networks by pruning away connections (Zhou et al., 2019; Ramanujan et al., 2020), while this work studies a different extraction mechanism, i.e., affine linear transformations.

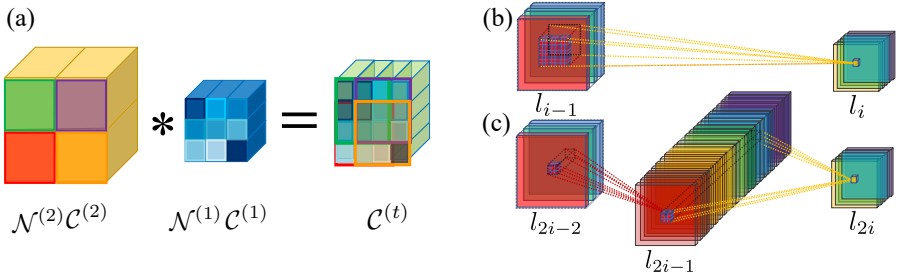

Figure 1: Construction idea. (a) The convolution of two tensors creates a larger tensor that represents a target tensor. (b) Target 2d-convolutional layer. Each channel is represented by a colored cuboid. A highlighted region visualizes a filter. (c) Two layers with random weights approximate a target layer.

## 2 UNIVERSAL APPROXIMATION

Standard convolutional neural networks (CNNs) and fully-connected feed-forward neural networks, which can be regarded as a special case of CNNs, possess the universal approximation property. This means that they can approximate any continuous function if they are wide and deep enough Gühring et al. (2022); Shen et al. (2022). Our main objective is to show how convolutional random CNNs with trainable affine linear transformations like BatchNorm inherit this ability.

Our argument is of the following structure. We want to show that a source network $f_s$ with relatively arbitrary weight tensors can be adapted with affine linear feature transformations to solve a general task of interest. Let us assume that a target network $f_t$ is given that solves such a task. This target exists because of the universal function approximation property of neural networks. Given that target and the source weights, we thus have to find affine linear transformations parameterized by $\gamma$ so that the source approximates the target $f_s(x \mid \gamma) \approx f_t(x)$. To make these statements more precise, we introduce a more rigorous mathematical notation.

### 2.1 BACKGROUND AND NOTATION

Our expressiveness results hold for general convolutional structures with or without skip connections. A convolutional neural network $f : \mathcal{D} \subset \mathbb{R}^{c_0 \times d_0} \to \mathbb{R}^{c_L \times d_L}$ is defined on a compact domain $\mathcal{D}$ and has channels $\bar{c} = [c_0, c_1, ..., c_L]$, i.e., depth $L$ and width $c_l$ in layer $l \in [L] := \{0, ..., L\}$. Without loss of generality, let $\mathcal{D} \subset [-1, 1]^{c_0 \times d_0}$ be contained in an interval (which could always be achieved by rescaling the input). A layer $l$ is generally parameterized by a a bias vector $\boldsymbol{b}^{(l)} \in \mathbb{R}^{c_l}$ and a weight tensor $\mathbf{W}^{(l)} \in \mathbb{R}^{c_l \times c_{l-1} \times k_l}$, which comprises the filters (or convolutional kernels).

$$\boldsymbol{x}_i^{(l)} = \phi\left(\boldsymbol{h}_i^{(l)}\right), \quad \boldsymbol{h}_i^{(l)} = \mathcal{C}^{(l)}(\boldsymbol{x}^{(l-1)}) = \sum_{j=1}^{c_{l-1}} \mathbf{W}_{ij}^{(l)} * \boldsymbol{x}_j^{(l-1)} + b_i^{(l)}, \tag{1}$$

$\boldsymbol{h}^{(l)}$ is called pre-activation and is transformed by an univariate activation function $\phi(x)$. To simplify and generalize our notation, we have flattened the filter dimension to $k_l$. In case of the common 2d convolutions, the weight tensor would actually have the size $\mathbf{W}^{(l)} \in \mathbb{R}^{c_l \times c_{l-1} \times k'_{1,l} \times k'_{2,l}}$ so that $k_l = \dim(k'_{1,l}, k'_{2,l})$. The convolution operation between any 2-dimensional tensors $K$ and $X$ is defined as $(\boldsymbol{K} * \boldsymbol{X})_{ij} = \sum_{i',j'} K_{i'j'} X_{(i-i'+1)(j-j'+1)}$ in this case. We assume that the inputs are always suitably padded with zeros and that the symbol $*$ performs the convolutions in the right dimensions (for simplicity with stride 1). The flattened notation just makes it easier to discuss higher dimensional filters at the same time. The type of convolution determines the dimension of the effective filter $\dim(k_1, k_2)$ that describes the composition of two consecutive filters with dimensions $k_1$ and $k_2$. For instance, the composition of a $k'_{1,1} \times k'_{1,2}$ filter with a $k'_{2,1} \times k'_{2,2}$ results in a filter with dimension $\dim(k_1, k_2) = (k'_{1,1} + k'_{2,1} - 1) \times (k'_{2,1} + k'_{2,2} - 1)$.

In addition to convolutional layers, we also allow for residual and more general skip connections. Skip connections modify the network above as $\boldsymbol{x}_i^{(l)} = \phi\left(\boldsymbol{h}_i^{(l)}\right) + \sum_{t=1}^{l-1} \sum_{j=1}^{c_t} \boldsymbol{S}_{ij}^{(l,t)} * \boldsymbol{x}_j^{(t)}$. We could accommodate this general form but this is not required for universal function approximation.

In residual architectures, which are often employed to improve the trainability of very deep CNNs, usually, only one or two layers are skipped so that $\boldsymbol{S}^{(l,t)}$ is nonzero for $t = l + 1$ or $t = l + 2$.

In addition, most modern architectures consist of additional normalization layers $\mathcal{N}(x)$ that tend to substantially improve the training and generalization performance of the model. They are either positioned before the activation function and after the convolution operation $\phi\left(\mathcal{N}\left(\boldsymbol{h}_i^{(l)}\right)\right)$ or immediately after the activation function $\mathcal{N}\left(\phi\left(\boldsymbol{h}_i^{(l)}\right)\right)$ in case of a post-activation structure. As pre-activations usually lead to better performance, at least for Batch Normalization (BN) Ioffe & Szegedy (2015) in ResNets He et al. (2015b), we focus our analysis on this case. However, our general approach would also transfer to post-activations. Most normalization strategies apply an (affine) linear transformation of their input vector. Examples include BN, one of the most effective and frequently deployed approaches. Yet, it is highly memory intensive, since it works best in combination with relatively high batch sizes. Other variants like weight normalization (Salimans & Kingma, 2016; Huang et al., 2017), weight standardization (Qiao et al., 2019), instance normalization (Ulyanov et al., 2016), Instance Enhancement Batch Normalization (Liang et al., 2020), or switch normalization (Luo et al., 2019) can address this issue and still achieve competitive performance in several cases (Brock et al., 2021b;a). They all can be formulated effectively as $\mathcal{N}(\boldsymbol{h}) := \gamma\boldsymbol{h} + \beta$, where each channel is transformed separately. $\gamma$, $\beta \in \mathbb{R}^{c_l}$ are learnable parameters in most cases. The shift is not always tuned but crucial to avoid singularites, in which neurons get switched off completely (Qiao et al., 2019). $\beta$ thus replaces the bias parameter of the previous convolutional layer, which is often removed by a centering operation. $\gamma$ and $\beta$ both have components that are estimated based on parameter Salimans & Kingma (2016) or batch statistics Ioffe & Szegedy (2015), which are implicitly integrated in the above formulation. Regarding them explicitly would not change our theoretical results.

Note that in standard neural network architectures, $\gamma$, $\beta$ do not contribute to the expressiveness of the architectures and thus the functions that the networks can potentially represent, as they could be integrated into the weight tensor and bias of the previous convolutional layer. Thus, $\gamma$, $\beta$ contribute to the overparameterization of the networks. Their main role is to change the training dynamics and help avoid exploding weight norms or switching off neurons by accident.

Interestingly, our results together with (Frankle et al., 2021; Giannou et al., 2023) indicate that the normalization parameters might not just control the range of activations but might significantly contribute to learning meaningful representations, especially in the presence of high width and/or depth overparameterization. Given a convolutional layer, $\gamma$, $\beta$ define linear combinations and shifts of pre-activation features that give the architecture full expressive power if sufficiently many features are available. As we prove, even if the weight tensor and the biases remain frozen in their initial random state and are not trained, learning only the normalization parameters $\gamma$, $\beta$ allows the network to represent any target layer of a certain size.

## 2.2 Two convolutional layers for one target layer

Our main goal is to derive a potentially random convolutional source network structure $\mathcal{S}$ that is able to represent arbitrary target convolutional layers $\mathcal{C}^{(t)}$ of a given dimension only with the help of affine transformations. This implies that the affine transformations of the normalization layers of $\mathcal{S}$ are fully expressive. Our main construction approach represents each convolutional target layer by the composition of two convolutional source layers $\mathcal{S} = \mathcal{N}^{(2)}\mathcal{C}^{(2)}\mathcal{N}^{(1)}\mathcal{C}^{(1)}$ as visualized in Fig. 1. For simplicity, let us first assume that the activation functions of the first source layer are the identity, while the activation functions that follow the target layer and the second source layer coincide. Later, we will also discuss how to adapt our results to almost arbitrary activation functions of the first layer.

For each possible target with parameters $(w_{ijk}^{(t)}, b_i^{(t)})$, we have to find source parameters $(\gamma_i^{(2)}, \beta_i^{(2)}, \gamma_J^{(1)}, \beta_J^{(1)})$ for $i \in [c_2], J \in [c_1], j \in [c_0]$ so that $w_{ijk}^{(t)} = \gamma_i^{(2)} \sum_J \sum_{(p,q)\in\mathcal{I}(k)} w_{iJp}^{(2)} \gamma_J^{(1)} w_{Jjq}^{(2)}$, where the kernel indices $p, q \in \mathcal{I}(k)$ correspond to the position $k$ in the tensor that is formed by the composition $\mathcal{C}^{(2)}\mathcal{C}^{(1)}$ (see Fig. 1 (a) for an example). The biases fulfill $\beta_J^{(1)} = 0$ for linear activation functions and $\beta_i^{(2)} = b_i^{(t)}$. While this system of equations is quadratic, we can reduce it to a linear one in $\gamma^{(1)}$ by solving for $\gamma_i^{(2)}$ first. For each $i \in [c_2]$, we

choose a pivotal element $(j_i, k_i) = \text{argmax}_{j,k}|w_{ijk}^{(t)}|$ and use $\gamma_i^{(2)}$ to make sure that $w_{ij_ik_i}^{(t)}$ is exactly represented. Note that we could pick any element $(ijk)$ with nonzero $|w_{ijk}^{(t)}|$. We simply pick the pivotal one for numerical stability. This transforms our problem to

$$\mathbf{M}\gamma^{(1)} = \mathbf{0}, \text{ for } M = (m_{IJ}) \text{ with } m_{IJ} := \sum_{(p,q)\in\mathcal{I}(k)} w_{iJp}^{(2)}w_{Jjq}^{(1)} - \frac{w_{ijk}^{(t)}}{w_{ij_ik_i}^{(t)}} \sum_{(p_i,q_i)\in\mathcal{I}(k_i)} w_{iJp_i}^{(2)}w_{Jj_iq_i}^{(1)} \tag{2}$$

that we have to solve for nontrivial $\gamma^{(1)} \neq \mathbf{0}$. The index $I$ corresponds to the position of the index set $(i, j, k)$ in a flattened representation, while $J$ denotes the index of $\gamma_J^{(l)}$ in that same representation. The following theorem states the conditions when this system of linear equations has a solution.

**Lemma 2.1** (Single layer reconstruction). *Let two consecutive convolutional and normalization layers $\mathcal{S}(\mathbf{x}) = \mathcal{N}^{(2)}\left(\mathcal{C}^{(2)}\left(\mathcal{N}^{(1)}\left(\mathcal{C}^{(1)}(\mathbf{x})\right)\right)\right)$ of channel dimensions $c_2$, $c_1$, $c_0$ and kernel dimensions $k_2$, $k_1$ be given. Then, each target convolutional layer $\mathcal{C}^{(t)}$ of dimensions $c_2$, $c_0$, and $k_t$ with nonzero output channels $\sum_{j,k}|w_{ijk}^{(t)}| > 0$ for all $i \in [c_2]$ can be represented exactly by adjusting $\mathcal{N}^{(1)}$ and $\mathcal{N}^{(2)}$ if $\mathbf{M} = (m_{IJ})$ as defined by Eq. (2) has nullspace dimension $\ker M \geq 1$.*

**Can all target networks be represented this way?** Importantly, the matrix $\mathbf{M}$ with $c_2c_0\dim(k_2, k_1) - c_2$ rows depends on the target, which has $c_2c_0k_t$ degrees of freedom. Thus, if we picked the target network dependent on the source $\mathcal{S}$, we could reduce the dimension of $\mathbf{M}$'s column space by 1, for instance, if $k_1 = 1$ and $w_{ijk}^{(t)}/w_{ij_ik_i}^{(t)} = w_{i1p}^{(2)}w_{1jq}^{(1)}/(w_{i1p_i}^{(2)}w_{1j_iq_i}^{(1)})$, which can only be fulfilled if the right hand side's magnitude is bounded by 1. We therefore have to incorporate an additional assumption on the target network to translate this insight into a bound on the source width. In practice, this excludes an irrelevant target space, as we will also demonstrate in experiments.

**Theorem 2.2** (Minimum width requirement). *Assume that the tensor entries of $\mathcal{S}$ are drawn independently from the target and mutually independently of each other from continuous probability distributions with finite mean and nonzero variance. Then Problem (2) has a nontrivial solution with high probability if (i) $\Delta k = dim(k_2, k_1) - k_t \geq 0$ and (ii) $c_1 \geq c_2c_0dim(k_2, k_1) - c_2 + 1$.*

This result also covers fully-connected layers, where all channel dimensions are one-dimensional $k_2 = k_1 = k_t = 1$. In contrast to Giannou et al. (2023), we only require a width of $c_1 \geq c_2c_0 - c_2 + 1$ instead of $c_2c_0$. The reason is that we also use the scale parameters $\gamma^{(2)}$ of the second layer for representing the target. Yet, we also have to exclude pathological targets in this statement. The maximum width requirement to really represent any possible target network can be derived by setting $\gamma^{(2)}$ to arbitrary nonzero values, which makes $\mathbf{M}$ independent of the target network itself so that it becomes generally invertible.

**Theorem 2.3** (Maximum width requirement). *Let the tensor entries of $\mathcal{S}$ in Theorem 2.1 be drawn independently from continuous probability distributions with finite mean and finite nonzero variance. Then $\mathcal{S}$ can represent any nonzero target with high probability if (i) $\Delta k = dim(k_2, k_1) - k_t \geq 0$ and (ii) $c_1 \geq c_2c_0dim(k_2, k_1)$.*

Note that (ii) implies that our width requirement depends on how well our source kernel dimensions align with the kernel dimension of the target. Our flattened representation of the kernel dimension hides the fact that we need the kernel of the target layer to be fully covered by the convolution of the two source layers. If additional elements are produced by the product that are not present in the target kernel, we need more parameters to guarantee that we can learn a representation of the target that sets such elements to zero. Thus, ideally, we consider convolutional layers with $\dim(k_2, k_1) = k_t$ to minimize our width requirement in (i). This can be achieved most easily by choosing one of the convolutional source layers to be one-dimensional, e.g., $k_1 = 1$ and the other layer to have identical kernel dimensions as the target layer, i.e., $k_2 = k_t$. To simplify notations in the following, we assume that $k_2 = k_t$ and skip the index $k_1 = 1$.

**Residual blocks.** While we have focused our exposition on convolutional source layers, Frankle et al. (2021) have primarily analyzed residual source blocks, as they could train deeper architectures more easily. Our results can be easily adapted to this case still assuming a convolutional target layer. The

adaptation depends on the position of the second normalization layer relative to the skip connections. In the simpler case, we have $\mathcal{S} = \mathcal{N}^{(2)}\mathcal{C}^{(2)}\mathcal{N}^{(1)}\mathcal{C}^{(1)} + \mathcal{C}^{(skip)}$. Applying our previous results to the modified target $W^{(t)} - W^{(skip)}$, where $W^{(skip)}$ is defined so that it is compatible with $W^{(t)}$, would suffice. In the more common case of an outer normalization $\mathcal{S} = \mathcal{N}^{(2)}\left[\mathcal{C}^{(2)}\mathcal{N}^{(1)}\mathcal{C}^{(1)} + \mathcal{C}^{(skip)}\right]$, we have to solve a modified system of linear equations that relies on the same matrix $\mathbf{M}$ but does not solve for $\mathbf{0}$ but a vector $\mathbf{c}$ with

$$\mathbf{M}\gamma^{(1)} = \mathbf{c}, \text{ for } c_I = \frac{w_{ijk}^{(t)}}{w_{ij_ik_i}^{(t)}} w_{ij}^{(skip)}. \tag{3}$$

## 2.3 ALMOST ARBITRARY ACTIVATION FUNCTIONS

Previously, we have assumed that the first layer of the source network has the identity as activation function, which made the biases of $\mathcal{N}^{(1)}$ obsolete. Yet, most common neural network architectures employ non-linear activation functions after every layer. To explain the experimental findings by (Frankle et al., 2021), we therefore have to cover at least ReLUs $\phi(x) = \max\{x, 0\}$ after the first source layer. As we show, many activation functions enable an approximate reconstruction of a target layer by modifying our results for linear activation functions. We just need to be able to scale and shift the bounded input so that it is approximately linearly transformed by $\phi$, as visualized in Fig. 2.

**Theorem 2.4.** *Assume that the conditions of Lemma 2.1 are met by $\hat{\mathcal{S}}$. Let another source $\mathcal{S}$ be defined as $\mathcal{S}(\mathbf{x}) = \mathcal{N}^{(2)}\left(\mathcal{C}^{(2)}\left(\phi\left(\mathcal{N}^{(1)}\left(\mathcal{C}^{(1)}(\mathbf{x})\right)\right)\right)\right)$ with the same convolutional layers as $\tilde{\mathcal{S}}$ but potentially different normalization layers. Based on the normalization layers of $\tilde{\mathcal{S}}$, define two constants $Q_1 := \max_J |\tilde{\gamma}_J^{(1)}| \sum_{jk} |w_{Jjk}^{(1)}|$ and $Q_2 := \max_{i,k} |\tilde{\gamma}_i^{(2)}| \sum_J |w_{iJk}^{(2)}|$.*

*If for a given $\epsilon > 0$ there exist constants $\gamma, b, m, c \in \mathbb{R}$ so that $|\phi(\gamma x + b) - (m\gamma x + mb + c)| \leq \epsilon\gamma m/Q_2$ for all $x \in [-Q_1, Q_1]$, then the normalization layers can be adjusted such that $|\mathcal{S}(\mathbf{x}) - \mathcal{C}^{(t)}(\mathbf{x})| \leq \epsilon$ for all $\mathbf{x} \in [-1, 1]^{c_0}$, where $|\cdot|$ denotes the supremum norm.*

The proof is presented in the appendix. Essentially, this theorem states that we can transfer our previous results to nonlinear activation functions as long as we find a region of the activation function that is approximately linear and the error scales sublinearly. Several relevant activation functions have this property. For instance, ReLUs $\phi(x) = \max\{x, 0\}$ and leakyReLUs $\phi(x) = \max\{x, 0\} - \alpha\max\{-x, 0\}$ inflict zero error with $\gamma = 1$, $b = Q_1$, $m = 1$, $c = 0$, and thus even work with $\epsilon = 0$. $\phi(x) = \tanh(x)$ fulfills $|x - \phi(x)| \leq x^3/3$ for $|x| \leq 1$. Thus, the choice $\gamma = \min\{(3\epsilon/Q_2)^{1/2}, 1/Q_1\}$, $b = 0$, $m = 1$, $c = 0$ would be sufficient. Similarly, sigmoids $\phi(x) = (\tanh(x/2) + 1)/2$ work with $\gamma = \min\{(12\epsilon/Q_2)^{1/2}, 1/Q_1\}$, $b = 0$, $m = 1/4$, $c = 1/2$. Thus, our theory attests that the experimental evidence for ReLUs Frankle et al. (2021) and theoretical results for ReLUs on fully-connected networks (Giannou et al., 2023) also transfer to other activation functions.

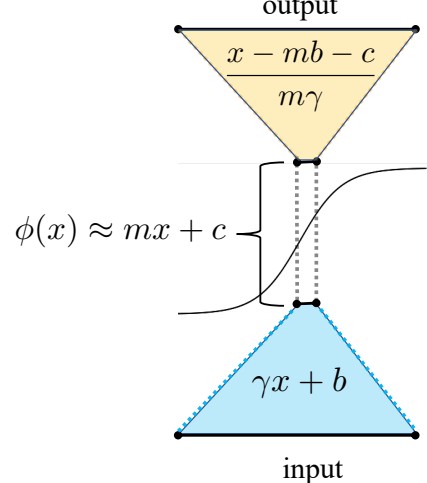

Figure 2: Approximation of the identity. After linear transformation of an input, the activation function $\phi(x)$ is evaluated in an approximately linear region. Another linear transformations maps the output back to the original input.

## 2.4 TRADE-OFF BETWEEN WIDTH AND DEPTH

So far we have established representation results for training only the normalization layers of randomly initialized weight tensors for different activation functions, where we would approximate one target layer by exactly two source layers.

**Multiple target layers.** Similarly to lottery ticket existence proofs (Burkholz, 2022b; da Cunha et al., 2022), our results for the representation of a single target layer can also be transferred to the composition of multiple target layers by utilizing two consecutive layers in the source network to approximate a target layer. In consequence, the source network would have double the depth of a

potential target network. As long as each layer inflicts an error of maximally $O(\epsilon/L)$, where $L$ is the depth of the network, under mild assumptions on the target (Burkholz, 2022b; da Cunha et al., 2022) the overall approximation error can be bounded below $\epsilon > 0$.

**Implications of 2-layer construction.** The width $c_1$ of the first source layer in our construction is very large in comparison with the input and output dimensions. To achieve full expressiveness, it has to scale quadratically in the input and output, i.e., $c_1 \propto c_2 c_0 k_2 k_1$, which corresponds to the dimension of a potential target tensor and thus the trainable parameters. Even if a high proportion of the network parameters is set to random values, we still need to ensure that we have a sufficient number of degrees of freedom. Thus the number of trainable parameters in the normalization layer cannot be reduced magically. Only if the target tensor and the random weight would align well, could we expect to require fewer parameters. The experiments by Frankle et al. (2021), however, were conducted with an architecture that consists of multiple layers of more similar size. Yet, the authors observed that roughly $1/3$ of output neurons was learned to be switched off.

**Why can it be beneficial to switch off neurons?** According to our theoretical results, removing neurons in the output layer has the effect that the number of channels $c_1$ in the first source layer could reach the required with to represent a target output layer. Alternatively creating additional random outputs would negatively influence the representation, as random combinations would contribute to each output of the next layer. Yet, switching off neurons is not the only option to effectively utilize the available degrees of freedom.

**Utilizing multiple source layers.** Instead of using two source layers with very wide hidden dimension, also more source layers of lower (and thus more balanced) width can be employed to represent a single target layer. This can be shown trivially for residual source networks that have at least twice the width of the input channel dimension, as a target tensor can be represented as composition of multiple residual layers that concatenate the input $\mathbf{x}$ and a current state of the target tensor construction as $y = \mathbf{x} + \sum_i \mathbf{W_i x}$ (see (Giannou et al., 2023) for a construction for fully-connected source and target networks). The downside of this representation is that it requires a higher number of total trainable parameters (at least by a factor of $4$) than a representation based on 2 layers. Our next next result for convolutional targets and source networks establishes a much stronger width requirement.

**Theorem 2.5** (Deeper source networks). *Let the the source network $\mathcal{S}$ consist of $L$ consecutive random convolutional layers (with linear activation functions). Then the parameters of the normalization layers can be adjusted with high probability such that a given target convolutional layer can be represented exactly if (i) $\Delta k = dim(k_1, ..., k_L) - k_t \geq 0$ and (ii) $\sum_{l=1}^{L} c_l \geq c_L c_0 dim(k_1, ..., k_L) + L$ and (iii) $c_l \geq c_0$ for all source layers $l \in [L]$.*

In conclusion, to utilize lower source widths $c_i$, we have to pay with $L$ more trainable parameters. $L$ degrees of freedom are lost as the parameters $\gamma^l$ of different layers are scale invariant. The main construction idea of the proof is to show inductively that we can decompose the problem into a linear system of equations and a system that is polynomial in fewer parameters $\gamma_i^l$ than the original problem.

**Implications.** While it is remarkable that learning normalization layers is fully expressive, our theoretical results study primarily the conditions that allow us to represent almost arbitrary target networks of a given size and thus imply strong width requirements. In practice, however, specific random tensor distributions that fit to a learning task might possess a relevant inductive bias that allows the training of smaller random networks. Understanding the nature of promising random distributions could be a viable strategy to effectively reduce the number of required trainable parameters in future.

## 3 EXPERIMENTS

### 3.1 RECONSTRUCTION ALGORITHMS

To reconstruct specific target tensors in our experiments based on random source networks, we have to solve large scale systems of linear equations. While there exist numerical algebra solvers such as the pytorch function *torch.linalg.solve*, they suffer from accumulating approximation errors for larger matrices. To reduce the error, we finetune the resulting solutions with LBFGS as implemented by the Pytorch function *torch.optim.LBFGS* that minimizes the mean squared error between our constructed network and the target. The error can still be sufficiently high to hamper exact reconstruction (see Tables 1 and 2). Interestingly, solving an overparameterized linear system with LBFGS that focuses

on solving $\gamma_1$ and fixes $\gamma_2$ to almost arbitrary values reduces the approximation error greatly and enables near perfect reconstruction.

## 3.2 PROOF OF PRINCIPLE: BATCH NORMALIZATION IS SUFFICIENT

The goal of our experiments is to verify the validity of our construction and provide empirical evidence for the fact that random features allow us to reconstruct a target network with exactly the same number of trainable parameters as we need to represent the target. All experiments were conducted on a machine with Intel(R) Core(TM) i9-10850K CPU @ 3.60GHz processor and GPU NVIDIA GeForce RTX 3080 Ti and we report averages and 95%-confidence intervals for 3 independent repetitions. To

Table 1: Target reconstruction for CIFAR10. Averages and 0.95 standard confidence intervals are reported for 3 independent source network initializations. Columns correspond to different random feature distributions and rows to networks with different activation functions.

|  | TARGET | HE NORMAL | HE UNIFORM | ORTHOGONAL | SPARSE HE NORMAL |
|---|---|---|---|---|---|
| RELU | 94.12 | $94.1 \pm 0.08$ | $94.0 \pm 0.1$ | $94.12 \pm 0.07$ | $94.12 \pm 0.07$ |
| LRELU | 94.37 | $94.34 \pm 0.05$ | $94.1 \pm 0.1$ | $94.4 \pm 0.2$ | $94.3 \pm 0.1$ |
| TANH | 93.66 | $93.5 \pm 0.1$ | $93.6 \pm 0.1$ | $93.45 \pm 0.05$ | $93.5 \pm 0.1$ |
| SIGMOID | 92.9 | $92.8 \pm 0.04$ | $92.5 \pm 0.1$ | $92.7 \pm 0.1$ | $92.8 \pm 0.04$ |

obtain suitable targets with different activation functions, we have trained VGG18-like structures that consists of 18 layers (17 convolutional with 3-dimensional filters, one final linear layer) of channel width $64$ in the intermediary layers with 592657 parameters in total. We consider two standard image classifcation benchmark datasets, CIFAR10 and CIFAR100 (Krizhevsky, 2009) and use a standard training procedure: SGD with 5 warmup epochs and linear learning rate increase, followed by 200 epochs of cosine annealing with initial learning rate 0.1.

Our targets have only width $c = 64$, as this enables us to repeatedly solve the related large scale systems of linear equations with memory and time intensive specialized numerical linear algebra methods (as implemented by Pytorchs torch.linalg.solve function). These equations arise when we explicitly match the targets according to the construction of our theorems. As the number of rows of the involved matrix $M$ scales as $c^2 k_t - c + 1 = 64^2 9 - 64 + 1 = 36801$, we are still able to match target networks with different activation functions. Note that our target performance still exceeds the accuracy of the networks which resulted from training only BN parameters of wide and deep ResNet or VGG architectures (Frankle et al., 2021). While previous experimental work was focused on ReLUs, we demonstrate that the same principles apply also to other activation functions. Specifically, we report results for ReLU, LeakyReLU, tanh, and sigmoid.

To verify the practical relevance of our theoretical claims, we explicitly construct scale and shift parameters of networks with random weights so that the source network with adapted affine linear transformations matches a given target network. To achieve that, we consider source networks where two layers represent one target layer as stated in Theorem 2.2, where the first layer has kernel dimension $1$. The source weight tensors have been randomly sampled from standard distributions that are commonly applied to initialize neural networks before training. We consider normal or uniform distributions as proposed by He et al. (He et al., 2015a), random orthogonal weight tensors (Pennington et al., 2017; Burkholz & Dubatovka, 2019), and a sparse He normal distribution, where each weight tensor element is set to $0$ with probability $0.5$ (Liu et al., 2021).

Table 2: Target reconstruction for CIFAR100. (See Table 1 for a detailed description.)

|  | TARGET | HE NORMAL | HE UNIFORM | ORTHOGONAL | SPARSE HE NORMAL |
|---|---|---|---|---|---|
| RELU | 70.46 | $70.34 \pm 0.02$ | $70.1 \pm 0.2$ | $70.3 \pm 0.2$ | $70.2 \pm 0.2$ |
| LRELU | 70.22 | $94.37 \pm 0.01$ | $93.88 \pm 0.16$ | $94.12 \pm 0.07$ | $94.12 \pm 0.07$ |
| TANH | 69.65 | $69.5 \pm 0.03$ | $69.3 \pm 0.2$ | $69.4 \pm 0.1$ | $69.6 \pm 0.05$ |
| SIGMOID | 67.42 | $94.1 \pm 0.08$ | $93.88 \pm 0.16$ | $94.12 \pm 0.07$ | $94.12 \pm 0.07$ |

Note that the random weight matrix $\tilde{M}$, which is required to have full rank, meets this requirement. However, the approximation error for problems of this size is high enough to induce small performance variations in the reconstruction as visible in Tables 1 and 2. This problem is not substantially amplified for larger targets. As a proof of concept, we have additionally obtained target networks with width $c = 100$ and ReLUs that reach an accuracy of $94.94$ and $73.34$ on CIFAR10 and CIFAR100, respectively. For He normal random weights, we our performance after reconstruction is $94.93 \pm 0.03$ and $73.37 \pm 0.05$. Altogether, our experiments verify our theory and target construction approach.

We could also attain a higher reconstruction accuracy if we were willing to further increase the width of the source network above the required minimum, as our gradient based solution algorithms benefit greatly from this small amount of overparameterization in terms of learning speed and accuracy, as has also been observed in the context of learning deep neural networks. If we increase the width by less than 80 or 120 parameters per layer, we do not observe any performance degradation if we employ overparameterized LBFGS (a variant of gradient descent) to minimize the mean squared error between our construction and the target.

**Adjusting only BN parameters is computationally feasible.** From a conceptual point of view, we have thus presented not only a theoretical but also practical existence proof that training only BN parameters is sufficient, the required models are of feasible size and can still outperform the results that were obtained by learning only BN parameters from scratch Frankle et al. (2021).

**Training from scratch misses the mark.** Frankle et al. (2021) have conducted experiments with standard convolutional and residual architectures on common image benchmark data, yet, could not achieve the expected performance of similar architectures. A possible explanation might be that the considered architectures were not designed based on the derived based on the presented insights and thus were neither sufficiently large nor structurally promising.

To test this hypothesis, we have trained a convolutional architecture that would theoretically correspond to a VGG18-like architecture with hidden width 100 on CIFAR10. Each target layer could be represented by two random source networks with linear activation function after the first layer. This choice avoids potential issues with switched off ReLUs after the first layer that could result from unstable learning dynamics. Yet, even if we provide it with additional overparameterization (like $10^4$ additional channels) and extended warmup cycles, training the normalization parameters only achieves a generalization error of $90.35 \pm 0.1$ in three independent runs. These are clearly not competitive with the generalization accuracy of potential target models (of at least 94%).

Training normalization parameters alone seems to require different specialized learning schedules. The future development of such algorithms could positively affect the learning dynamics of normalization layers also when they are trained in combination with the remaining network parameters.

## 4 CONCLUSIONS

Our work shows that normalization layers of modern convolutional neural network architectures are fully expressive if the weight tensors are sufficiently wide or deep and stay fixed to their initial random values. This rigorously proves the conjecture by (Frankle et al., 2021) and derives the precise width and depth requirements for a random source network relative to a target network that enable the exact reconstruction of a target network. In comparison with previous experimental results, we have extended the theoretical and experimental evidence to a relatively general class of activation functions (that includes but is not limited to ReLUs) and different random tensor distributions.

Based on the derived target reconstruction, we have provided an empirical proof of principle that normalization layers alone can achieve the expected performance on standard benchmark data, which was previously not attainable by training the normalization layers from scratch with standard learning schedules (Frankle et al., 2021).

This evidence complements insights into the diverse roles or normalization layers, in particular BN layers. As they add redundant parameters, normalization is primarily believed to impact the training dynamics positively, for instance, by enabling larger learning rates. The derived expressiveness of normalization layers suggests, however, that they could also contribute in a representational sense by learning linear combinations of currently available features.

ACKNOWLEDGEMENTS

We gratefully acknowledge funding from the European Research Council (ERC) under the Horizon Europe Framework Programme (HORIZON) for proposal number 101116395 SPARSE-ML.

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

## A    APPENDIX

### A.1    PROOF OF LEMMA 2.1

*Statement* (Lemma 2.1 in main manuscript). Let two consecutive convolutional and normalization layers $\mathcal{S}(\mathbf{x}) = \mathcal{N}^{(2)}\left(\mathcal{C}^{(2)}\left(\mathcal{N}^{(1)}\left(\mathcal{C}^{(1)}(\mathbf{x})\right)\right)\right)$ of channel dimensions $c_2$, $c_1$, $c_0$ and kernel dimensions $k_2$, $k_1$ be given. Then, each target convolutional layer $\mathcal{C}^{(t)}$ of dimensions $c_2$, $c_0$, and $k_t$ with nonzero output channels $\sum_{j,k}|w_{ijk}^{(t)}| > 0$ for all $i \in [c_2]$ can be represented exactly by adjusting $\mathcal{N}^{(1)}$ and $\mathcal{N}^{(2)}$ if $\mathbf{M} = (m_{IJ})$ as defined by Eq. (2) has nullspace dimension $\ker M \geq 1$.

*Proof.* For each possible target with parameters $(w_{ijk}^{(t)}, b_i^{(t)})$, we have to find source parameters $(\gamma_i^{(2)}, \beta_i^{(2)}, \gamma_J^{(1)}, \beta_J^{(1)})$ for $i \in [c_2], J \in [c_1], j \in [c_0]$ so that

$$w_{ijk}^{(t)} = \gamma_i^{(2)} \sum_J \sum_{(p,q)\in\mathcal{I}(k)} w_{iJp}^{(2)} \gamma_J^{(1)} w_{Jjq}^{(2)}, \tag{4}$$

where the kernel indices $p$ and $q$ correspond to the position $k$ in the tensor that is formed by $\mathcal{C}^{(2)}\mathcal{C}^{(1)}$ (see Fig. 1 (a) for an example). The biases fulfill $\beta_J^{(1)} = 0$ for linear activation functions and $\beta_i^{(2)} = \beta_i^{(t)}$.

While this system of equations is quadratic, we can reduce it to a linear one in $\gamma^{(1)}$ by solving for $\gamma_i^{(2)}$ first. For each $i \in [c_2]$, we choose a pivotal element $(j_i, k_i) = \operatorname{argmax}_{j,k}|w_{ijk}^{(t)}|$ and use $\gamma_i^{(2)}$ to make sure that $w_{ij_ik_i}^{(t)}$ is exactly represented.

This results in

$$\gamma_i^{(2)} = \frac{w_{ij_ik_i}^{(t)}}{\sum_J \sum_{(p_i,q_i)\in\mathcal{I}(k_i)} w_{iJp_i}^{(2)} \gamma_J^{(1)} w_{Jj_iq_i}^{(2)}}. \tag{5}$$

Replacing this expression in Eq (4) leads to the stated definition of $\mathbf{M}$. □

## A.2 Proof of Thm. 2.2

*Statement* (Thm. 2.2 in main manuscript). Assume that the tensor entries of $\mathcal{S}$ are drawn independently from the target and mutually independently of each other from continuous probability distributions with finite mean and nonzero variance. Then Problem (2) has a nontrivial solution with high probability if (i) $\Delta k = \dim(k_2, k_1) - k_t \geq 0$ and (ii) $c_1 \geq c_2 c_0 \dim(k_2, k_1) - c_2 + 1$.

*Proof.* Overall, we have to account for $c_2 c_0 k_t$ degrees of freedom of the target. Condition (i) ensures that the kernel dimension of our construction matches the kernel dimension of our target network. If $\Delta k > 0$, we obtain additional constraints, as the constructed kernel also needs to assign 0 to values that do not overlap with the target kernel. Overall, we therefore have $c_2 c_0 \dim(k_2, k_1)$ degrees of freedom. $c_2$ of those are automatically fulfilled by our choice of $\gamma_2$ that results in matrix entries $m_{IJ} := \sum_{(p,q) \in \mathcal{I}(k)} w_{iJp}^{(2)} w_{Jjq}^{(1)} - \frac{w_{ijk}^{(t)}}{\sum_{(p_i, q_i) \in \mathcal{I}(k_i)}} w_{ij_i k_i}^{(t)} w_{iJp_i}^{(2)} w_{Jj_i q_i}^{(1)}$. Yet, we need an additional column to ensure that the nullspace dimension is $\ker \mathbf{M} \geq 1$ such that a nontrivial solution $\gamma^{(1)}$ exists to the respective system of linear equations. $\mathbf{M}$ therefore needs to have at least $c_1 \geq \tilde{c} = c_2 c_0 \dim(k_2, k_1) - c_2 + 1$ columns.

To construct a solution, we effectively add another condition to our set of equations, namely $\sum_{d=1}^{\tilde{c}} \gamma_d^{(1)} = 1$ that demands a non-trivial $\gamma^{(1)}$ and invert the respective matrix.

What is left to show is that a quadratic submatrix $\mathbf{M}_s$ of $\mathbf{M}$ with $\tilde{c} - 1$ rows and $\tilde{c} - 1$ columns has full rank and is thus invertible with high probability. In this case, a possible solution for our construction would fulfill $\gamma_s^{(1)} = \mathbf{M}_s^{-1} \mathbf{z}$, where the vector $\mathbf{z}$ is defined via a column of $\mathbf{M}$ that is not included in the submatrix. Without loss of generality, let this column be the first one and $\mathbf{M}_s = (m_{IJ})_{I,J \in \{2,...,\tilde{c}\}}$ so that $\mathbf{z} = -\mathbf{M}_{:1}$ and $\gamma_1^{(1)} = 1 - \sum_{d=2}^{\tilde{c}} \gamma_d^{(1)}$.

Our proof concludes when we can show that $\mathbf{M}_s$ has a full rank with high probability. To do so, we employ a similar strategy as Giannou et al. (2023), who show that the Khatri-Rao product of two random matrices is full rank with high probability. Therefore, we also use the following Lemma 5 of (Giannou et al., 2023).

**Lemma A.1** (Lemma 5 of (Giannou et al., 2023)). *Let $p(\mathbf{x})$ be a polynomial of degree $d$, $\mathbf{x} \in \mathbb{R}^n$. If $p$ is not the zero polynomial, then the set $\mathcal{S} := \{\mathbf{x} \in \mathbb{R}^n \mid p(\mathbf{x}) = 0\}$ is or measure zero (with respect to the Lebesgue measure).*

As a consequence, it is sufficient to show that $\det \tilde{\mathbf{M}}_s$ is nonzero for a specific assignment $\tilde{M}$ of the otherwise random tensor elements. As this would imply that $\det \mathbf{M}_s$ is not the zero polynomial, the set $\mathcal{S}$ of source weight tensors that lead to non-invertible $\mathbf{M}_s$ with $\det \mathbf{M}_s = 0$ has zero probability measure.

A specific assignment that we consider for this purpose has elements $\tilde{w}_{iJp_i}^{(2)} = 0$ and $\tilde{w}_{iJp}^{(2)} = \delta_{ii_J} \delta_{pp_J}$ and $w_{Jjq}^{(1)} = \delta_{jj_J} \delta_{qq_J}$ for one $(p, q) \in \mathcal{I}(k)$, where $\delta$ stands for the Kronecker delta. Consequently, $\tilde{m}_{IJ}$ simplifies to the identity matrix $(\tilde{m}_{IJ}) = (\delta_{IJ})$ Thus, we have $\det \tilde{\mathbf{M}}_s = 1 \neq 0$ and random $M_s$ are invertible with high probability. $\qquad\square$

## A.3 Proof of Thm. 2.3

*Statement* (Thm. 2.3 in main manuscript). Let the tensor entries of $\mathcal{S}$ in Theorem 2.1 be drawn independently from continuous probability distributions with finite mean and finite nonzero variance. Then $\mathcal{S}$ can represent any nonzero target with high probability if (i) $\Delta k = \dim(k_2, k_1) - k_t \geq 0$ and (ii) $c_1 \geq c_2 c_0 \dim(k_2, k_1)$.

*Proof.* The proof of this theorem is essentially already contained in the previous one. (ii) ensures that we can set $\gamma_i^{(2)} = 1$ and still solve the resulting system of linear equations $\mathbf{M}\gamma^{(1)} = \mathbf{w_t}, \mathbf{I}$ with $c_1 \geq c_2 c_0 \dim(k_2, k_1)$ rows and columns, where $m_{IJ} = \sum_{(p,q) \in \mathcal{I}(k)} w_{iJp}^{(2)} w_{Jjq}^{(1)}$.

We have already shown in the last proof that also a matrix with structure $\mathbf{M} = \sum_{(p,q) \in \mathcal{I}(k)} (w_{iJp}^{(2)} w_{Jjq}^{(1)})$ is invertible with high probability. We can simply use again the assign-

ment $\tilde{w}_{iJp}^{(2)} = \delta_{ii_J}\delta_{pp_J}$ for a $(p,q) \in \mathcal{I}(k)$ and $w_{Jjq}^{(1)} = \delta_{jj_J}\delta_{qq_J}$ to construct the identity matrix, which has nonzero determinant. □

## A.4   PROOF OF THM. 2.4

*Statement* (Thm. 2.4 in main manuscript). Assume that the conditions of Lemma 2.1 are met by $\hat{\mathcal{S}}$. Let another source $\mathcal{S}$ be defined as $\mathcal{S}(\mathbf{x}) = \mathcal{N}^{(2)}\left(\mathcal{C}^{(2)}\left(\phi\left(\mathcal{N}^{(1)}\left(\mathcal{C}^{(1)}(\mathbf{x})\right)\right)\right)\right)$ with the same convolutional layers as $\tilde{\mathcal{S}}$ but potentially different normalization layers. Based on the normalization layers of $\tilde{\mathcal{S}}$, define two constants $Q_1 := \max_J |\tilde{\gamma}_J^{(1)}| \sum_{jk} |w_{Jjk}^{(1)}|$ and $Q_2 := \max_{i,k} |\tilde{\gamma}_i^{(2)}| \sum_J |w_{iJk}^{(2)}|$.

If for a given $\epsilon > 0$ there exist constants $\gamma, b, m, c \in \mathbb{R}$ so that $|\phi(\gamma x + b) - (m\gamma x + mb + c)| \le \epsilon\gamma m/Q_2$ for all $x \in [-Q_1, Q_1]$, then the normalization layers can be adjusted such that $|\mathcal{S}(\mathbf{x}) - \mathcal{C}^{(t)}(\mathbf{x})| \le \epsilon$ for all $\mathbf{x} \in [-1,1]^{c_0}$, where $|\cdot|$ denotes the supremum norm.

*Proof.* We have to show that $|\mathcal{S}(\mathbf{x}) - \mathcal{C}^{(t)}(\mathbf{x})| \le \epsilon$ under the stated assumptions.

To do so, let us first define the normalization parameters of $\mathcal{S}$ that impose the required scaling so that we can approximate $\phi(x) \approx mx + c$.

$\gamma_J^{(1)} = \gamma\tilde{\gamma}_J^{(1)}$, $\beta_J^{(1)} = b$, $\gamma_i^{(2)} = \tilde{\gamma}_i^{(2)}/(m\gamma)$, and $\beta_J^{(2)} = \tilde{\beta^{(2)}}_J - \frac{mb+c}{m\gamma}$. If the activation function were $\phi_m(x) = mx + b$, then these normalization parameters would make $\mathcal{S}_m$ identical to $\tilde{\mathcal{S}}$ and thus also identical to the target layer $C_t = S_m$ according to Lemma 2.1.

With this definition, we get $\left|\phi\left(\mathcal{N}^{(1)}\left(\mathcal{C}^{(1)}(\mathbf{x})\right)\right)_J - m\gamma\left(\tilde{\mathcal{N}}^{(1)}\left(\mathcal{C}^{(1)}(\mathbf{x})\right)\right)_J - b\right| = \left|\phi\left(\gamma\tilde{\gamma}_J^{(1)}w_{J::}^{(1)} * \mathbf{x} + b\right)_J - m\gamma\tilde{\gamma}_J^{(1)}w_{J::}^{(1)} * \mathbf{x} + b\right| \le \epsilon\gamma m/Q_2$ based on our assumption on $\phi$, since the input $|\sum_J \tilde{\gamma}_J^{(1)}w_{J::}^{(1)} * \mathbf{x}| \le Q_1$.

It follows that

$$\left|\mathcal{S}(\mathbf{x}) - \mathcal{C}^{(t)}(\mathbf{x})\right| = |\mathcal{S}(\mathbf{x}) - \mathcal{S}_m(\mathbf{x})| \le \max_{i,\mathbf{x}} |\mathcal{S}(\mathbf{x})_i - \mathcal{S}_m(\mathbf{x})_i| \tag{6}$$

$$= \max_{i,\mathbf{x}} \left|\gamma_i^{(2)} \sum_{J,p} w_{iJp}^{(2)} \left[\phi\left(\mathcal{N}^{(1)}\left(\mathcal{C}^{(1)}(\mathbf{x})\right)\right)_J - m\gamma\left(\tilde{\mathcal{N}}^{(1)}\left(\mathcal{C}^{(1)}\right)\right)_J - b\right]\right| \tag{7}$$

$$\le \frac{Q_2}{m\gamma} \max_{J,\mathbf{x}} \left|\phi\left(\mathcal{N}^{(1)}\left(\mathcal{C}^{(1)}(\mathbf{x})\right)\right)_J - m\gamma\left(\tilde{\mathcal{N}}^{(1)}\left(\mathcal{C}^{(1)}\right)\right)_J - b\right| \le \frac{Q_2}{m\gamma}\epsilon\gamma\frac{m}{Q_2} \tag{8}$$

$$= \epsilon \tag{9}$$

□

## A.5   PROOF OF THM. 2.5

*Statement* (Thm. 2.5 in main manuscript). Let the the source network $\mathcal{S}$ consist of $L$ consecutive random convolutional layers (with linear activation functions). Then the parameters of the normalization layers can be adjusted with high probability such that a given target convolutional layer can be represented exactly if (i) $\Delta k = \dim(k_1, ..., k_L) - k_t \ge 0$ and (ii) $\sum_{l=1}^{L} c_l \ge c_L c_0 \dim(k_1, ..., k_L) + L$ and (iii) $c_l \ge c_0$ for all source layers $l \in [L]$.

*Proof.* We will prove this statement by mathematical induction. The base case is covered by Thm. 2.2 with two source networks.

We want to show that for any target network can solve with high probability

$$\sum_{l=1}^{L} \sum_{I_l=1}^{c_l} \prod_{l=1}^{L} w_{I_l I_{l-1} q_l}^{(l)} \gamma_{I_l}^{(l)} = w_{t, I_L I_0 k}. \tag{10}$$

As in the derivation of Thm. 2.2, we can always eliminate the outer $\gamma_{I_L}^{(L)}$ and reformulate our problem

$$m_{IJ} = \prod_{l=1}^{L} w_{I_l I_{l-1} q_l}^{(l)} - \frac{w_{t,J}}{w_{t,J_L^*}} w_{J_L^* I_{L-1} q_L^*}^{(L)} \prod_{l=2}^{L-1} w_{I_l I_{l-1} q_l}^{(l)} w_{J_1^* I_{l-1} q_1^*}^{(1)} \tag{11}$$

such that

$$\sum_{l=1}^{L-1} \sum_{I_l=1}^{c_l} m_{IJ} \prod_{l=1}^{L-1} \gamma_{I_l}^{(l)} = 0 \tag{12}$$

with respect to a tensor $\mathbf{M} = (m_{IJ})$ that depends on a flattened index set with indicees $I$ and $J$ that correspond to index combinations $(I_1, ..., I_L, q_1, ..., q_L)$ and pivotal elements $w_{t,J_L}$ whose indicees are signified with $*$.

**Induction hypothesis:** Assume that $L - 1$ layers can represent any target network of the stated dimensions in the theorem. We furthermore assume that

$$\sum_{l=1}^{L-1} \sum_{I_l=1}^{c_l} m_{IJ} \prod_{l=1}^{L-1} \gamma_{I_l}^{(l)} = 0 \tag{13}$$

can be solved for a full rank $\mathbf{M}$.

**Induction step:** Our goal is to show that there exists a nontrivial solution to the larger system

$$\sum_{l=1}^{L} \gamma_{I_L}^{(L)} \sum_{I_l=1}^{c_l} m_{I_L I J} \prod_{l=1}^{L-1} \gamma_{I_l}^{(l)} = 0 \tag{14}$$

To do that, we derive a tensor of reduced rank $\mathbf{B} = (b_{IJ})$ such that

$$\sum_{I_L=1}^{c_L} \gamma_{I_L}^{(L)} b_{I_L J} = 0 \tag{15}$$

is solvable and

$$b_{I_L J} = \sum_{I_L, I} m_{I_L I J} \prod_{l=1}^{L-1} \gamma_{I_l}^{(l)}. \tag{16}$$

Essentially, $B$ helps us to decompose the original problem in two solvable smaller problems. The first one is linear and therefore feasible if $B$ has full rank and the second one is covered by the induction hypothesis.

To derive $\mathbf{B}$, let us distinguish two complementary subsets of all possible $\tilde{c} = \sum_{l=1}^{L} c_l$ constraints. These index subsets are defined as $S$ and $S_c$ so that $S \bigcup S_c = [\tilde{c}]$. The sizes correspond to the last layer width $|S| = c_L$ and the number of equations that can be solved based on our induction hypothesis, i.e., $|S_c| = \tilde{c} - c_L$.

Solving Eq. (15) is only possible if the nullspace $\ker(B) \geq 1$ has a nonzero dimension. Furthermore, solving only $c_L$ equations must imply that the remaining $\tilde{c} - c_L$ equations are automatically fulfilled. Thus, we require the columns of $B$ to be linearly dependent. Concretely, there must exist scalars $\lambda_{s_c s}$ such that

$$b_{I_L s_c} = \sum_{s \in S} \lambda_{s_c s} b_{I_L s}.$$

Combining this with Eq. (16), we obtain the following system of equations

$$\sum_{I} m_{I_L I s_c} \prod_{l=1}^{L-1} \gamma_{I_l}^{(l)} = \sum_{s \in S} \lambda_{s_c s} \sum_{I} m_{I_L I s} \prod_{l=1}^{L-1} \gamma_{I_l}^{(l)} \tag{17}$$

or, equivalently,

$$\sum_{I} \left( m_{I_L I s_c} - \sum_{s \in S} \lambda_{s_c s} m_{I_L I s} \right) \prod_{l=1}^{L-1} \gamma_{I_l}^{(l)} = 0. \tag{18}$$

If there exist constants $g_{Is_c} = \left(m_{I_L Is_c} - \sum_{s \in S} \lambda_{s_c s} m_{I_L Is}\right)$, then the equation above becomes solvable according to the induction hypothesis. But do there exist $\lambda_{ss_c}$ such that the right hand side varies only $I$ and $s_c$? In fact, $\lambda_{ss_c}$ also solves a linear system of equations with exactly $c_L(\tilde{c} - c_L)$ constraints corresponding to the indicees $I_L$ and $s_c$. This can be easily derived by eliminating $g_{Is_c}$, for instance, by establishing $m_{\tilde{I}_L Is_c} - m_{I_L Is_c} = \sum_s \lambda_{ss_c}(m_{\tilde{I}_L Is} - m_{I_L Is})$. Solving this system leads us to define $g_{Is_c}$, which enables us to solve Eq. 18 based on the induction hypothesis. Given $\prod_{l=1}^{L-1} \gamma_{I_l}^{(l)}$, we have also identified $\mathbf{B}$ based on Eq. 16 and can finally solve Eq. 15 for $\sum_{I_L=1}^{c_L} \gamma_{I_L}^{(L)}$.

Finally, note that all of relevant matrices are invertible, since they have sufficient random elements. This can also be shown rigorously as in the other proofs by finding an exemplary assignment to the weight tensors that is invertible.

We thus have shown that we can solve the induction step, which concludes the proof. $\qquad\square$

