# OpenReview forum: "Batch normalization is sufficient for universal function approximation in CNNs"
_ICLR.cc/2024/Conference — ICLR 2024 poster_

### Official Review · Reviewer_9YBa · 2023-10-15

**Soundness:** 3 good
**Presentation:** 3 good
**Contribution:** 3 good
**Rating:** 8
**Confidence:** 3

**Summary:**

This paper explores the role of normalization techniques, particularly Batch Normalization (BN), in deep convolutional neural networks (CNNs). The authors provide a theoretical analysis to demonstrate that training normalization layers alone is adequate for universal function approximation, assuming a sufficient number of random features. This result applies to various CNN architectures, including those with or without residual connections and different activation functions, such as ReLUs. The authors also explain how this theory can elucidate the depth-width trade-off in network design and the empirical observation that disabling neurons can be beneficial.

**Strengths:**

- The paper provides a solid theoretical foundation for its claims, offering mathematical proofs and a well-structured argument.

- This paper is written in a clear and easily comprehensible manner, making it easy for readers to follow.

**Weaknesses:**

See Questions.

**Questions:**

- In the introduction, I'm a bit confused about the normalization being explored in this paper. The title mentions batch normalization, but the statement "we delve into the role of layer normalization" suggests layer normalization. Is this a typo?

- I'm puzzled by the assertion in Section 2 regarding the existence of $f_t$, which the author claims is due to the universal function approximation. Perhaps the author meant to refer to the Universal Approximation Theorem. However, such approximations in neural networks are typically conditional. Has the author considered these conditions? Generally, these conditions are not mild and are idealized. Does this affect the theory presented in this paper? The author should provide an explanation and discussion on this.

- I'm not entirely sure why throughout the paper, normalization is reduced to just a linear transformation and shift, i.e., $\gamma \mathbf{h} + \beta$. This includes batch norm, layer norm, instance norm, etc. They all have this form, so is their significance solely in the learnable parameters $\gamma$ and $\beta$? Of course, I understand that $[(x - \mu) / \sigma] \times \gamma + \beta$ can be equivalent to $\gamma \mathbf{h} + \beta$" in the end, but why emphasize batch normalization in the title? Where does "batch" come into play?

- Can the analysis apply to the existing advanced batch normalization improvements like IEBN [1] and SwitchNorm [2]. These missing works should be considered and added to the related works or analysis.

- The author needs to clarify the above questions. If these issues are addressed, I will consider these clarifications along with feedback from other reviewers in deciding whether to raise my score.


[1] Instance Enhancement Batch Normalization: An Adaptive Regulator of Batch Noise, AAAI

[2] Differentiable Learning-to-Normalize via Switchable Normalization, ICLR

---

> ### Author Response · Authors · 2023-11-21
>
> We thank the reviewer for the constructive feedback. In the following, we address their questions and concerns.
>
> - Indeed, we have used layer normalization when we meant normalization layers. We have revised the manuscript accordingly.
>
> - Universal Approximation Theorems often have mild assumptions on the activation functions (e.g. that they are monotonously increasing and continuous) and the smoothness of the data generating function (usually that it is continuous) [1].
> Depending on the considered architecture (e.g. a three layer neural network or a width-constrained perceptron with ReLUs, the approximation quality of the neural network is derived dependent on the number variable network parameters (e.g. the width of the 3-layer neural network or the depth of the width constrained perceptron) [2].
>
>     * By formulating theorems with respect to a given target network that solves a task of interest, we are flexible in which universal function approximation theorem could be invoked. Our results would inherit the specific assumptions.
>     * Also irrespective of the specific universal function approximation theorem, we can cover typical application cases, for which target networks have been obtained by training the networks in a classical way. We have clarified this argument on page 3.
>
> - Our title refers to batch normalization, as it is one of the first and thus most well known normalization techniques. For that reason, several theoretical investigations have focused on understanding its contribution to successful deep learning. Our title embeds our work in this rich line of research but we would be happy to change the it upon request to highlight the more general scope of our insights.
>
> - The analysis can be extended to IEBN and SwitchNorm. We thank the reviewer for pointing us these relevant references, which we have included as examples in our background section.
>
> [1] Gühring, Raslan, Kutyniok (2022). Expressivity of Deep Neural Networks. In P. Grohs & G. Kutyniok (Eds.), Mathematical Aspects of Deep Learning (pp. 149-199). Cambridge: Cambridge University Press. doi:10.1017/9781009025096.004
>
> [2] Shen, Yang, Zhang (2022). Optimal approximation rate of ReLU networks in terms of width and depth, Journal de Mathématiques Pures et Appliquées, Volume 157, Pages 101-135, ISSN 0021-7824.

---

> ### Comment · Reviewer_9YBa · 2023-11-23
> **Thank you for your response.**
>
> Thank you for your detailed response. Upon reflecting on your response, I still find myself puzzled about whether normalization can be analyzed solely through $\gamma \mathbf{h} + \beta$. While I understand that more intricate normalization sub-operations, such as "minus the mean, etc.", are challenging to analyze, I feel the need for a more reasonable explanation. Furthermore, I appreciate the contributions of this work and look forward to a well-founded explanation specifically addressing this issue. And I tend to raise my score if there are some rational explanations.

---

> > ### Author Response · Authors · 2023-11-23
> >
> > We sincerely thank the reviewer for their response and motivate below why the analysis of $\gamma \mathbf{h} + \beta$ is sufficient for our purpose. To understand this, we need to carefully distinguish between the role of normalization parameters for the trainability and the expressiveness of a neural network.
> >
> > **Trainability**
> > Normalization influences several aspects of the training dynamics. For instance, an important role is that it ensures that signal can propagate through neurons at least for some samples even when the network changes strongly due to high learning rates. To achieve this, the more intricate standardization operation is relevant. Effectively, it ensures reasonable values of $gamma$ and $\beta$ in $\gamma \mathbf{h} + \beta$, when the network is not well trained so that we do not have a lot of information how good values of $\gamma$ and $\beta$ could look like. Yet, the mean $\mu$ and $\sigma$ do not influence the actual expressiveness of the network, as they could just be integrated into $\gamma$ and $beta$.
> >
> > **Expressiveness**
> > In this work, we focus on the contribution of the normalization to the expressiveness of the network. We thus derive the values of the effective normalization parameters that would allow a network with random weights to approximate a target network. We could also separately define the mean $\mu$ and standard deviation $\sigma$ based on training data batches and the target network and adjust $\gamma$ and $\beta$ accordingly, but this would not change the principle and make the exposition more complicated without any need.
> >
> > Actually, we find the fact intriguing that affine linear transformations on the neuron level are already sufficient to achieve highly expressive neural network models. In fact, if we have linear activation functions, only the parameters $\gamma$ in the intermediary layer would be enough, as they can implement linear combinations of the features that are randomly transformed by the network weights and biases. The bias term $\beta$ in the intermediary source layer is primarily needed to shift the input into a linear region of the activation function. The bias $\beta$ of the second layer assumes the role of the original target bias (and compensates for shifts in the intermediary layer).
> >
> > Does this explanation address the source of the reviewer's puzzlement? We would be happy to further elaborate on unclear aspects.

---

> > > ### Comment · Reviewer_9YBa · 2023-11-23
> > > **Thank you once again for your response.**
> > >
> > > Thank you once again for your response. While I'm not entirely satisfied with this explanation (It doesn't take away from the fact that this paper deserves to be accepted), I have gained valuable insights. I recognize the complexity of this issue, as I've been contemplating it for quite some time. Without simplifying it to $\gamma \mathbf{h} + \beta$, normalization becomes challenging to analyze. Additionally, you are correct in noting that $\gamma \mathbf{h} + \beta$ already possesses powerful expressive capabilities. I encourage you to incorporate this perspective into the article and cite the following papers with this perspective:
> > >
> > > [1] Stabilize deep resnet with a sharp scaling factor tau, arxiv
> > >
> > > [2] Rezero is all you need: Fast convergence at large depth, UAI
> > >
> > > [3] ScaleLong: Towards More Stable Training of Diffusion Model via Scaling Network Long Skip Connection, NeurIPS
> > >
> > > [4] Fixup initialization: Residual learning without normalization,ICLR
> > >
> > > [5] How to start training: The effect of initialization and architecture, NeurIPS
> > >
> > >
> > >
> > > .
> > >
> > >
> > > .
> > >
> > >
> > > Overall, I tend to raise my score, thank you again.

---

> > > > ### Author Response · Authors · 2023-11-23
> > > > **Thank you for the great references.**
> > > >
> > > > We thank the reviewer for their response and sharing these references. We will be happy to integrate them in our discussion that we will extend according to our exchange.
> > > >
> > > > We agree that we need to understand normalization in more depth and from multiple perspectives that include the standardization operation in future, as it is a fundamental aspect of deep learning.

---

### Official Review · Reviewer_pcEF · 2023-10-30

**Soundness:** 2 fair
**Presentation:** 2 fair
**Contribution:** 3 good
**Rating:** 3
**Confidence:** 2

**Summary:**

The paper shows that, under certain conditions, training just the batch normalization (BN) parameters is enough to make a CNN a universal function approximator.

**Strengths:**

The paper practices experimental method really well (has hypothesis, describes experiments and shows results).

**Weaknesses:**

The work describes BN as a subset of a Layer Normalization (LN).
I'd note that they are both normalization layers, but BN is NOT a type of LN.
(I'm effectively rejecting the paper because it needs to be rewritten based on this)

BN, LN, GroupNorm, etc are all normalization layers, the only difference is the dimension along which normalization is done; GroupNorm (https://arxiv.org/abs/1803.08494) paper has a pictorial depiction of this.
Furthermore, given BN aggregates statistics across samples, the neural network (NN) output of BN changes if the set of images changes; during training this makes a NN with BN a statistical operator. LN operates within a sample; the NN output does NOT change depending on input samples; during training this makes a NN with LN a function (not statistical). The Online Normalization paper (https://arxiv.org/abs/1905.05894) does a good job talking about this.




- Reference are broken (there are things like "Theorem ??" in the paper).
- Paper's experiments are really small scale for modern Deep learning leaving the reader wondering if they will scale.

**Questions:**

-

---

> ### Author Response · Authors · 2023-11-21
>
> We thank the reviewer for their constructive feedback and address their concerns below.
>
> * According the reviewers suggestions, we have revised the manuscript and replaced our misnomer layer normalization by normalization or normalization layer.
>
> * We have also addressed the broken Theorem link in the experiment section.
>
> * One of our main contributions is that we have derived the precise scaling requirements. While our experiments primarily serve the validation of our theoretical claims, note that our insights have allowed us to achieve much better performance than previous empirical work [1].
>     - However, as we discuss on page 7 in Section 3.1 and on page 8, we have to acknowledge that the precision of numerical solvers is limited in solving large scale linear systems of equations. Note that the matrix $M$ in our experiments has already a dimension of up to $90000 \times 90000$ roughly.
>     - In fact, our theoretical results imply that training only BN parameters for computational savings is not a practically reasonable approach if we want to maintain high expressiveness. Only inductive bias in the weight distributions (e.g. as obtained by foundation models) could enable training success despite a low number of degrees of freedom (see Lemma 2.1).
>
>
> [1] Frankle, Schwab, Morcos. Training BatchNorm and only BatchNorm: On the expressive power of random features in CNNs. ICLR, 2021.

---

### Official Review · Reviewer_Zja7 · 2023-11-02

**Soundness:** 3 good
**Presentation:** 3 good
**Contribution:** 3 good
**Rating:** 8
**Confidence:** 5

**Summary:**

They provide explanation for the normalization by proving that training normalization layers alone is already sufficient for universal function approximation if the number of available, potentially random features matches or exceeds the weight parameters of the target networks that can be expressed.

**Strengths:**

1- The effectiveness has been well supported by experiments.

2- Well organized and clearly written.

3- The paper is appropriately placed into contemporary literature.

**Weaknesses:**

I read the whole paper excluding the appendix, I acknowledge the importance of their study and appreciate the detailed information.

**Questions:**

N/A

---

> ### Author Response · Authors · 2023-11-21
> **Comment**
>
> We thank the reviewer for their positive assessment of our work.

---

### Official Review · Reviewer_DqFm · 2023-11-02

**Soundness:** 3 good
**Presentation:** 1 poor
**Contribution:** 2 fair
**Rating:** 5
**Confidence:** 2

**Summary:**

The paper provides a proof that a convolutional multi-layer perceptron with batch normalization is an universal function approximator even when only the batchnorm parameters are trainable and everything else is fixed to its random initialization.
The claim is proved by providing a practical construction for a number of common activation functions and parameter random distributions.

Minimum model widths and depths are provided for the task of reproducing an arbitrary convolutional MLP.

The approach is validated by experiments on image classification of the CIFAR10 and CIFAR100 datasets.

**Strengths:**

Interesting theoretical contribution.

**Weaknesses:**

Unclear practical relevance. The contribution is marginal compared to known results about approximation with models with random fixed parameters.

**Questions:**

N/A

---

> ### Author Response · Authors · 2023-11-21
> **Highlighting practical relevance and theoretical significance**
>
> We thank the reviewer for the time and effort they put into their review. To address their concern, in the following, we highlight the practical relevance and theoretical significance of our contributions.
>
> **Practical relevance:**
>
> - The practical relevance of our work is based on insights that can guide the design of neural network architectures, in which BN parameters contribute significantly to the expressiveness of the neural networks (like in WideResNets).
>
> - Concretely, we have derived architectural constraints and make specific proposals how training only BN paramters can achieve the same performance as training a target network.
> This way, we could outperform previous experiments with architectures, where only BN parameters were trained [1].
>
> - We have proven rigorously that a higher depth can compensate for a too narrow width. This is a relevant insight, in particular, since the random networks that were analyzed empirically were not wide enough.
>
> - Our results imply that training only BN parameters for computational savings is not practically relevant if we want to maintain high expressiveness. Only inductive bias in the weight distributions (e.g. as obtained by foundation models) could enable training success despite a low number of degrees of freedom (see Lemma 2.1).
> Note that this insight was not well supported even for fully-connected networks because not all trainable parameters were actually utilized in the construction [2].
>
> **Significance of theoretical contributions:**
>
> - Convolutional layers are an important building block of many contemporary neural network architectures. Therefore, we have to understand their intricacies if we want to gain theoretical insights that are of practical relevance and, up to our knowledge, we are the first to do so in the context of training only training only normalization parameters.
> (Note that our insights actually allow us to outperform previous purely empirical work on that matter and explain their findings.)
> - Convolutional architectures are challenging to analyze theoretically in our context, as the composition of multiple filters induces non-trivial equations that are quadratic (and not linear) in the trainable BN parameters.
> Furthermore, the composition of convolutional source filters has to overlap with a target filter, which induces additional constraints that do not occur in fully-connected layers.
> - Even in comparison to previous work on fully-connected layers [2], we present an improved bound on the source width, which utilizes all trainable parameters (which requires solving a quadratic instead of a linear system of equations).
> Note that this improved bound is crucial for an important insight into the limitations of the overall approach.
> If we want to maintain full expressiveness, we cannot use fewer parameters.
> This insight is not fully supported by previous work [2], which does not utilize all trainable parameters.
> - Furthermore, we cover almost arbitrary activation functions and not only linear ones or ReLUs.
> - Conceptually, most challenging and novel is the proof of Theorem 2.5 that solves the depth versus width trade-off in the source network.
>
> [1] Frankle, Schwab, Morcos. Training BatchNorm and only BatchNorm: On the expressive power of random features in CNNs. ICLR, 2021.
>
> [2] Giannou, Rajput, Papailiopoulos. The expressive power of tuning only the normalization layers. COLT, 2023.

---

### Author Response · Authors · 2023-11-21
**Summary**

We thank all reviewers for their efforts and the constructive feedback that led us to revise our manuscript.
If there remain any open questions, we would be happy to address them.

In summary, we have
* removed the misnomer of layer normalization,
* removed a broken Theorem link,
* extended our universal approximation argument,
* and added additional proposed examples of normalization techniques that are covered by our theory.

With our theoretical insights, we have derived the precise conditions under which training only normalization layer parameters would be sufficient to solve a task of interest. Unfortunately, our results state that there is no free lunch and we still require a high number of degrees of freedom to achieve high expressiveness. Yet, our insights allowed us to construct random neural networks with adjusted BN parameters that would outperform previous experimental results on standard benchmark datasets. As our derivations could guide the design of neural network architectures that enable normalization layers to contribute effectively to the expressiveness of neural networks, we believe that our contributions could be of high interest to the ICLR community.

For a more detailed description of our contributions, we refer to our answer for Reviewer DqFm.

---

### Meta-Review · Area_Chair_uw2i · 2023-12-11

**Metareview:**

In this paper the authors provide an investigation of normalization techniques, in particular Batch Normalization (BN), in order to demonstrate that neural networks solely trained on BN parameters may be a universal function approximator.  The authors provide a mathematical proof to this effect and demonstrate their result empirically on reconstruction tasks based on CIFAR-10 and CIFAR-100.

The reviewers commented positively on the theoretical foundation for the claims and the mathematical proofs as well as the clarity of presentation. The reviewers also commented that the title, and references to the type of normalization explored need to be seriously updated and revised. In the latest version of the paper, I do not see these revisions in the paper and agree with the sentiment that this paper must be revised. This paper will be conditionally accepted so long as the heavy revisions are made to the title to the description of the normalization methods.

**Justification For Why Not Higher Score:**

N/A

**Justification For Why Not Lower Score:**

Clear presentation of solid theoretical results.

---

### Decision · Program_Chairs · 2024-01-16

Accept (poster)